# Evidence needs and community involvement in policy decisions for vaccine-preventable diarrhoeal infections among children under the age of five years: Stakeholder engagement in Ethiopia, Kenya, and Malawi

Chikondi A. Mwendera[1]*, Mackwellings Phiri[2,3], Rahma Osman[4],
Shewit Weldegebriel[5], Beatrice Ongadi[4], Chisomo Msefula[2], Amha Mekasha[5],
Samuel Kariuki[4], Catherine Beavis[1], Jessica A. Fleming[6], Dan Hungerford[1],
Nigel Cunliffe[1], Deborah Nyirenda[3,7‡], Neil French[1‡], GHRG-GI Consortium[¶]

1 Department of Veterinary and Ecological Sciences, Institute of Infection, Veterinary and Ecological Sciences, University of Liverpool, Liverpool, Merseyside, United Kingdom, 2 Department of Microbiology, Kamuzu University of Health Sciences, Blantyre 3, Malawi, 3 Department of Social Science, Malawi Liverpool Wellcome Programme, Blantyre 3, Malawi, 4 Department of Microbiology Research, Kenya Medical Research Institute, Nairobi, Kenya, 5 Department of Paediatrics and Child Health, Addis Ababa University, College of Health Sciences, Addis Ababa, Ethiopia, 6 Center for Vaccine Innovation and Access, PATH, Seatle, Washington, United States of America, 7 Liverpool School of Tropical Medicine, Liverpool, England, United Kingdom

¶ Membership of the GHRG-GI Consortium is provided in the Acknowledgements.
‡ Joint Senior authors on this work.
* c.mwendera@liverpool.ac.uk, mwenderac@gmail.com

## Abstract

Diarrhoeal diseases are a leading cause of morbidity and mortality among children under five years in low- and middle-income countries (LMICs) with *Shigella* and Enterotoxigenic *Escherichia coli* (ETEC), both targets of ongoing vaccine development, playing major roles. To inform the introduction of vaccines against these pathogens in Ethiopia, Kenya, and Malawi, we assessed evidence needs for technical policy decisions and community engagement in national policy development processes. Using qualitative methods, we conducted 27 in-depth interviews (IDIs) with institutional stakeholders (ministries of health, researchers, and partners) and one focus group discussions (FGDs) in each country with community representatives. Data were collected between May 2023 to September 2024 and analysed thematically. Institutional stakeholders emphasised the need for robust, localised evidence on disease burden, vaccine efficacy, safety, cost-effectiveness, programmatic feasibility, alongside structured mechanisms to integrate evidence into policy. They noted the absence of diarrhoeal disease-specific technical working groups in their countries and advocated for their establishment. Meanwhile, community representatives expressed vaccine acceptance but stressed the importance of clear communication of information delivered by trusted health workers and leaders. They highlighted the need for comprehensive information on new vaccines, particularly

**Data availability statement:** "Due to the small sample sizes in each country and the sensitive nature of the views shared during interviews and focus group discussions, full transcripts cannot be made publicly available. Sharing complete narratives could risk participant identification, which would breach confidentiality agreements and informed consent. Participants did not consent to public data sharing or deposition in external repositories. To safeguard participant privacy, access to the full data is restricted to the research team. However, de-identified excerpts relevant to and illustrative of the study's findings are presented within the paper.".

**Funding:** This study was conducted under the overall research by the Global Health Research Group on Gastrointestinal Infections (GHRG-GI), whose Principal Investigators include NC, CM, AM, SK, and NF. The research was supported by the National Institute for Health and Care Research (NIHR) using UK aid from the UK Government to support global health research (award number NIHR133066). The funders had no role in study design, data collection and analysis, decision to publish, or preparation of the manuscript.

**Competing interests:** The authors have declared that no competing interests exist.

regarding their safety and potential side effects. While community involvement in national policy-making processes was deemed low, participants highlighted the value of meaningful engagement to ensure ownership and uptake. The findings highlight the importance of robust evidence generation, effective communication, and inclusive community engagement to facilitate the successful introduction of *Shigella* and ETEC vaccines. Addressing these gaps can accelerate adoption, improve readiness, and foster trust in vaccination programmes.

## Introduction

Diarrhoeal diseases cause approximately 500,000 annual deaths among children under five years, with the heaviest burden falling on low- and middle-income countries (LMICs), particularly in sub-Saharan Africa (SSA) [1,2]. In SSA, rotavirus, *Shigella*, and Enterotoxigenic *Escherichia coli* (ETEC) are leading pathogens, fuelled by persistent gaps in water, sanitation, and hygiene (WASH) infrastructure [2–4]. While oral rehydration therapy and intravenous fluids for severe cases of diarrhoea, and antibiotics for *Shigella* and ETEC remain key treatments, their effectiveness is limited by poor access and antimicrobial resistance (AMR) [5–8]. Vaccines offer critical protection, as demonstrated by the success of rotavirus vaccine (RVV) programmes in Malawi, Ethiopia, and Kenya, which reduced rotavirus-associated hospitalisations, following their introduction in 2012, 2013, and 2014 respectively after the 2009 World Health Organization (WHO) recommendation [9–14]. However, modest RVV effectiveness in LMICs and uneven vaccine coverage persist, leaving rotavirus a leading cause of childhood diarrhoea [15,16].

Beyond rotavirus, *Shigella* and ETEC are major contributors to childhood diarrhoeal diseases in LMICs [17,18]. The rising burden of these pathogens, compounded by AMR, has prompted the WHO to prioritise the development of vaccines against them [19,20]. However, timely vaccine introduction is critical. For instance, since the WHO recommendation, the RVV introduction took three, four, and five years in Malawi, Ethiopia, and Kenya, respectively [10,11,13,14]. Such delays, more common in LMICs than high-income countries [21], represent missed opportunities for community health benefits and hinders progress towards global targets, including the immunisation-related UN Sustainable Development Goals (SDGs) [22]. Addressing these delays requires governments to ensure that decisions on vaccine introduction are impartial, timely and evidence-based [23,24]. Generating robust local evidence to inform policy decisions on new vaccines in therefore, critical. The importance of such evidence has been demonstrated by the International Centre for Diarrhoeal Disease Research, Bangladesh (icddr,b), which has contributed over six decade of research shaping global diarrheal disease control from developing oral rehydration solution to demonstrating the feasibility and cost-effectiveness of the oral cholera vaccine delivery through routine health systems in high-risk urban setting [25,26].

Community engagement is proven to strengthen health interventions, particularly during implementation [27,28]. However, meaningful participation in national

policy-making processes in Africa is often limited by tokenistic consultations, power imbalances, and weak decentralisation [29,30]. Bridging these gaps requires integrating communities into policy design (not just during implementation), to warrant sustainable and equitable health outcomes.

To ensure readiness for future *Shigella* and ETEC vaccines, LMICs must strengthen preparation, while improving RVV coverage. As part of the NIHR Global Health Research Group on Gastrointestinal Infections (GHRG-GIs), we conducted in-depth interviews (IDIs) with institutional stakeholders (ISs) in Ethiopia, Kenya, and Malawi to assess evidence needs for *Shigella* and ETEC vaccine introduction. This work aligns with the group's broader efforts to generate evidence, build capacity, and engage communities for childhood enteric vaccines in eastern and southern Africa. Additionally, focus group discussions (FGDs) with community engagement groups gathered local perspectives on community participation in vaccine policy processes and strategies to foster community ownership and acceptance.

## Methodology

### Ethics statement

Ethical approvals for the study were obtained from the respective ethics review boards in each country: Department of Pediatrics at Addis Ababa University (030–23-ped) in Ethiopia, Kenya Medical Research Institute's Scientific Ethical Review Unit (SERU NO: KEMRI/SERU/CMR/P00221-010–2022/4637) in Kenya, and the Kamuzu University of Health Sciences (P.10/22/3790) in Malawi. Additionally, ethical approval was granted by the University of Liverpool, UK (Reference number 12443). Prior to participating in the in-depth interviews (IDIs) and focus group discussions (FGDs), all participants were provided with comprehensive information about the study and given the opportunity to ask questions. Informed consent was obtained individually from each participant. Written consent was secured for all IDIs conducted in the three countries, as well as for FGDs in Kenya and Malawi. In Ethiopia, verbal consent was obtained for FGDs in accordance with local ethical approvals. Participants were also informed of their right to withdraw from the study at any time without any consequences. In addition, information related to ethical, cultural, and scientific considerations specific to inclusivity in global research is included in the supporting information (S2 Checklist). To ensure quality, we followed the consolidated criteria for reporting qualitative studies (COREQ) (S1 Checklist).

### Study design

This qualitative study employed a phenomenological approach [31], which seeks to explore and interpret individual's lived experiences of a given phenomenon. In this study, the approach was used to understand how institutional stakeholders (ISs) and community representatives (CPs) experienced and perceived their involvement in vaccine policy processes. Data were collected through IDIs with ISs and FGDs with CPs.

### Participants, recruitment, and setting

**Institutional stakeholders (ISs).** At national level, we purposively sampled 27 ISs across Ethiopia, Kenya, and Malawi supplemented by snowball sampling. Participants included representatives from ministries of health (MoHs), WHO country offices, national public health and technical advisory bodies, immunisation programme managers, researchers, and clinicians. Selection was guided by participants' institutional knowledge, decision-making roles, and relevant expertise. To protect confidentiality, IS characteristics are reported in aggregated categories (Table 1). Of the 27 ISs, seven were female, and their work experience in current roles ranged from over 5 to more than 20 years.

**Community participants (CPs).** CPs were drawn from the community engagement and involvement (CEI) groups established by the GHRG-GI in high-density urban areas with limited infrastructure and high infectious disease burdens. CEI members were community volunteers selected through a participatory process involving community nomination, open dialogue, and democratic selection. Criteria prioritised diversity, gender balance, and inclusion of caregivers, health

**Table 1. Participants characteristics.**

| Participant group | IDIs and FGDs conducted | Description |
|---|---|---|
| **Institutional stakeholders (ISs)** | 27 IDIs | Ministry of Health officials (n = 8); researchers from national or international research institutions (n = 9); representatives of NGOs and partner organisations (n = 10). |
| **Community participants (CPs)** | 3 FGDs | Community Engagement and Involvement (CEI) group members in Ethiopia (n = 10), Kenya (n = 10), and Malawi (n = 7). |

workers, teachers and marginalised groups affected by gastrointestinal infections, ensuring representation of diverse social, cultural, and professional backgrounds.

One FGD was conducted in each country: ten participants in Ethiopia (five females and five males), ten in Kenya (six females and four males), and seven in Malawi (four females and three males).

## Study sites

This study was part of the GHRG-GI in Ethiopia, Kenya, and Malawi, countries with a high burden of childhood diarrhoeal diseases and active immunisation programmes. Research was conducted in high-density urban areas with limited infrastructure and elevated infectious disease risk, identified in collaboration with national partners: Addis Ababa University (Ethiopia), the Kenya Medical Research Institute (KEMRI), and the Malawi-Liverpool Wellcome Trust (MLW).

In Ethiopia, the study was conducted at Tikur Anbessa Specialized Teaching Hospital in Addis Ababa, with CEI groups based at Teklehaimanot and Kirkos sub-cities, where urbanisation has contributed to overcrowding, inadequate clean water access, and high rates of infection [32]. In Kenya, the site was Mukuru Health Centre in Nairobi, targeting informal settlements in Kwa Reuben and Kwa Njenga, where poor water and sanitation are key drivers of diarrhoeal diseases [33]. In Malawi the study was conducted at Bangwe Health Centre in Blantyre's informal settlements, where HIV/AIDS, malaria, tuberculosis, and diarrhoeal diseases are prevalent amid inadequate WASH conditions [34].

**Data collection.** Data were collected between May 2023 and September 2024. In Ethiopia, the IDIs and FGDs were conducted from 21/05/2023–10/11/2023; in Kenya from 15/10/2023–20/11/2023 and again after ethical renewal from 25/02/2024–08/05/2024; and in Malawi, from 10/02/2024–05/09/2024. The IDIs were conducted using a topic guide (S1 Text) that was piloted in each country and lasted up to 60 minutes. The IDIs were conducted in English, as participants were national-level actors who routinely used English in their professional roles and were comfortable conversing in it. For this reason, the lead author (CM), an experienced qualitative researcher, facilitated these interviews directly. While in-person interviews were prioritised and conducted during field visits to each country, some participants were unavailable at those times due to professional commitments. To ensure their perspectives were not excluded, these interviews were conducted virtually at the participants' request. The FGDs with CPs were conducted in the usual CEI group meeting environments in each country to ensure participants felt comfortable and familiar with the setting. In Ethiopia, the FGD was held at Kirkos Health Centre (Addis Ababa); in Kenya at Mukuru Health Centre (Nairobi); and in Malawi at Bangwe Health Centre (Blantyre). All FGDs were facilitated by trained social scientists from the GHRG-GI country teams and co-authors, namely SW (Ethiopia), RO (Kenya), and (MP) Malawi. A topic guide (S1 Text) explored community awareness of vaccines policies related to childhood diarrhoeal infections and perceived influence in decision-making. FGDs were conducted in local languages (Amharic in Ethiopia, Swahili in Kenya, and Chichewa in Malawi) and lasted approximately 90 minutes.

Discussions were audio-recorded and observational notes taken to contextualise the depth. This was done following participant briefing and the acquisition of informed consent.

**Data management and analysis.** Audio recordings from FGDs were transcribed verbatim and translated into English, while IDIs were transcribed verbatim. The authors involved in data collection conducted quality checks to ensure transcript accuracy and integrity. Qualitative analysis was performed thematically in NVivo 12 [35], using both inductive and deductive approaches guided by Braun and Clarke's six-step framework [36]. Two authors (CM and MP) independently coded the FGD and IDI transcripts to develop initial coding frameworks, which were then applied consistently across all transcripts. Codes were organised into categories, refined into subthemes, and synthesised into overarching themes for each dataset. Cross-country analysis identified shared perspectives through comparative synthesis of FGD and IDI data.

## Results

### Emerging themes

Four main themes related to evidence needs for technical decision-making and community engagement in national policy development processes emerged from the IDIs and the FGDs, including (1) policy involvement and evidence-related challenges, (2) evidence needs for policy decisions, (3) trusted sources of evidence, and (4) strategies for increasing evidence use. Sub-themes for each of these main emerging themes are outlined in Fig 1.

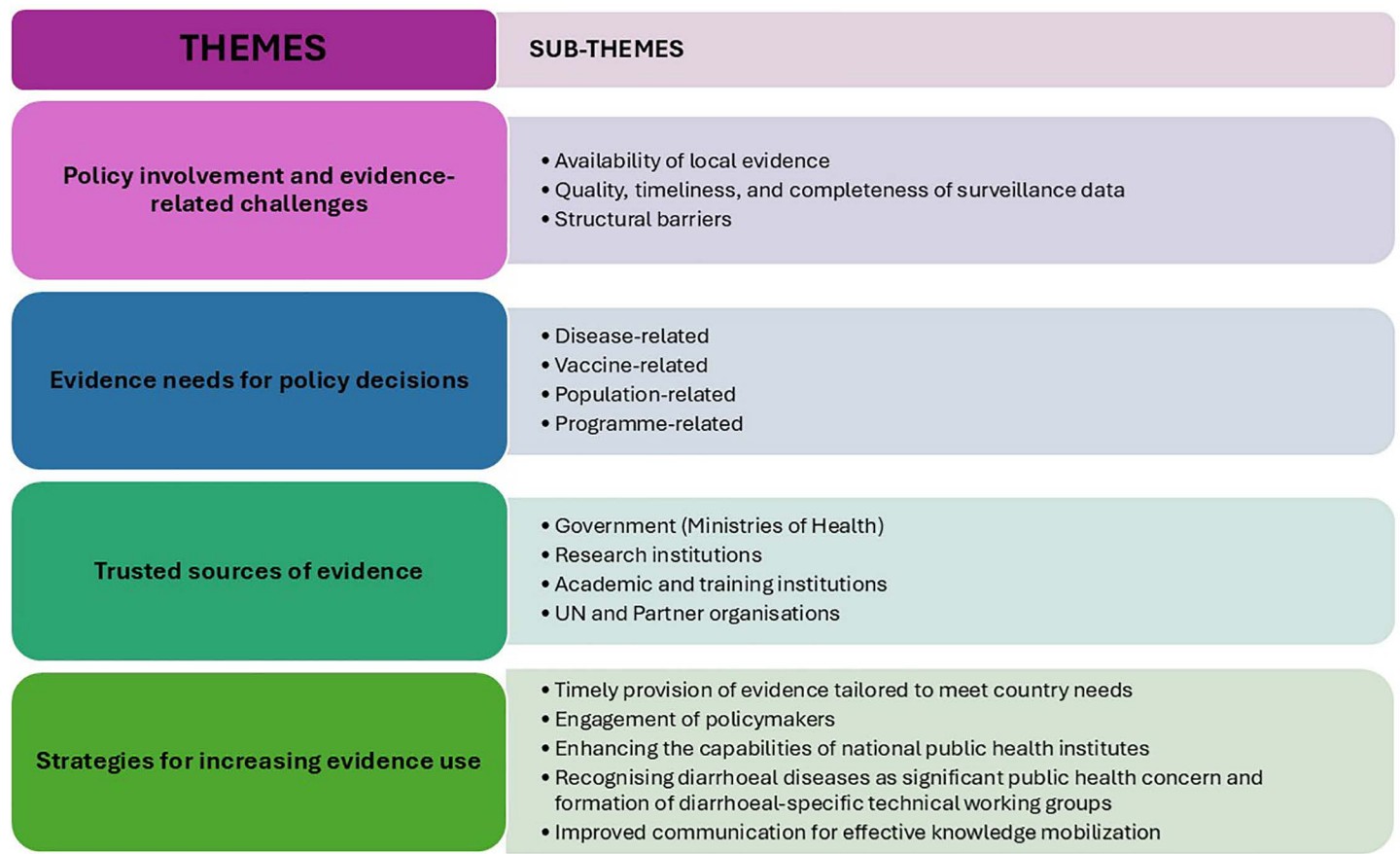

**Fig 1. Emerging themes and sub-themes on evidence needs for technical decision-making and community engagement in national policy development processes.**

## 1. Policy involvement and evidence-related challenges

ISs across Ethiopia, Kenya, and Malawi reported varying levels of involvement in vaccine policy processes, most often through technical working groups (TWGs) such as NITAGs, drafting policy documents, providing expert consultations, and conducting research to inform MoH policy decisions. For example, one Kenyan participant described:

*"I sit in the Kenya National Immunisation Technical Advisory Group (KENITAG), which advises the ministry of health on all matters related to vaccines, including modifying existing schedules or introduction of new vaccines".* (**Institutional Stakeholder in Kenya**)

Similarly, an Ethiopian stakeholder highlighted how surveillance data underpinned past decisions:

*"My centre was a diarrhoeal disease surveillance centre, so, we collected the surveillance data and based on that the burden of rotavirus as an important pathogen was confirmed. And that was strong evidence for the introduction of the rotavirus vaccine in this country".* (**Institutional Stakeholder in Ethiopia**)

While involvement was clear, ISs also underscored challenges in generating and applying evidence to guide timely vaccine adoption. A recurring concern was the lack of robust local disease burden data, which often delays decisions despite global WHO recommendations:

*"Actually, having local evidence is one of the challenges most of the time when you introduce a vaccine…. we need to have some local evidence on how it will be effective in Ethiopia, given that Ethiopia has its own specific contexts".* (**Institutional Stakeholder in Ethiopia**)

Beyond availability, ISs raised concerns about the quality, timeliness, and completeness of surveillance data, which undermined confidence in policy decisions. As one Malawian stakeholder noted:

*"When it comes to data, some of the challenges that we are facing are issues with the quality of the data, completeness, the timeliness of reporting…… I think we still have opportunities for improvement as we move along".* (**Institutional Stakeholder in Malawi**)

ISs also pointed to structural barriers, including the absence of diarrhoeal disease–specific TWGs within MoHs, meaning diarrhoeal evidence was not systematically reviewed alongside higher-profile diseases like HIV, malaria, or tuberculosis. This, combined with limited funding and dependence on external partners to generate cost-effectiveness or feasibility studies, further restricted the evidence base.

CPs, by contrast, reported little or no involvement in national policy processes. While some participated in local initiatives, most associated 'policy' only with implementation and vaccine uptake. Nevertheless, CPs advocated for greater inclusion, suggesting structured consultation with community leaders or mothers' groups to ensure that local perspectives on vaccines were represented in national decisions. These sentiments were expressed as below:

*"I have never been involved in any policy formulation process at national level, but I would say I took part at community level. I was called by the chief for our community to discuss about strategies for preventing diarrhoea and other water borne diseases in our community. I would say that I have attended such kind of meetings several times especially during the time when cholera was rampant here in Bangwe".* (**FGD participant in Malawi**)

*"To gain the entire community's idea, it is preferable if they involve the leaders of the community. It will be fantastic to be involved in policy formulation through community leaders or with groups of mothers who have children under 5".* (**FGD participant in Ethiopia**)

## 2. Evidence needs for policy decisions

ISs identified four key evidence domains (see Fig 2) as essential for informing policy decisions on the introduction of new childhood enteric vaccines into EPIs: disease and pathogen-related evidence, vaccine-related evidence, population-based evidence, and programme readiness and logistics.

### Disease and pathogen-related evidence

Across all three countries, ISs consistently highlighted the importance of robust local data on disease burden, including pathogen type, serotypes, morbidity, hospitalisation, and mortality. Such data were seen as the starting point for vaccine adoption, helping policymakers prioritise among competing health needs:

*"The main decision points are influenced by availability of data on the burden of disease. Basically, how many people get sick? How many of those also die? And you've got to compare it with other existing pathogens or other existing diseases."*

**(Institutional Stakeholder in Kenya)**

### Vaccine-related evidence

ISs highlighted vaccine efficacy, effectiveness, safety, and WHO recommendations as critical drivers of adoption, while Kenyan ISs also stressed affordability during GAVI transition:

**Disease and pathogen-related evidence**

- Type of diarrhoeal-causing pathogen
- Pathogen-specific morbidity and mortality
- Pathogen-specific hospitalization rates
- Pathogen serotypes and genotypes
- National disease distribution patterns
- Risk of infection

**Vaccine-related evidence**

- Vaccine safety
- Vaccine efficacy and effectiveness
- Duration of protection
- Vaccine cost-benefit analyses
- Vaccine administration

**Population-based evidence**

- Perception and acceptability of vaccines by both communities and caregivers
- The economic impact of diarrhoeal diseases on individual families
- Equity

**Programme readiness and logistics**

- Supply chain
- Workforce availability and capacity
- Resources and infrastructure

**Fig 2. Evidence domains to guide policy decisions.**

*"Information about the vaccine and disease is very important, for example, from the vaccine side, safety, effectiveness, and efficacy are quite important. So, we must be able to convince the regulatory bodies, you know, about safety and efficacy. And we get comfort when WHO approves."* **(Institutional Stakeholder in Ethiopia)**

*"So, we look at alternative costs… what are the alternatives and then we also weigh if there is treatment, which one is cheaper? Is it vaccines or treatment? So that in the end you're able to rationalise and see whether vaccines would be cheaper".* **(Institutional Stakeholder in Kenya)**

CPs, by contrast focused on practical vaccine characteristics and transparent information on safety, side effects, and administration schedules before accepting new vaccines. CPs in Ethiopia also drew on collective memory of diseases like smallpox and pertussis to affirm their trust in vaccines:

*"I won't accept a new vaccine unless I know how it works. We need information on its advantages, side effects, mode and age for administration".* **(FGD participant in Kenya)**

*"I don't think there would be obstacles. The community knows well about benefits of vaccines from experience. The people have learned from past harms caused by diseases like smallpox and pertussis, so they have been taking their children for vaccine services".* **(FGD participant in Ethiopia)**

**Population-based evidence**

ISs noted the importance of social and cultural considerations for equitable access, including strategies to ensure vaccines reach vulnerable populations. They also raised concerns about the accuracy of population denominators used in planning, which can distort vaccine coverage estimates:

*"The other thing is improving population estimates, especially to calculate targets and vaccination coverage. Different denominators are used when calculating vaccine needs versus vaccine coverage making it difficult to determine accurate targets."* **(Institutional Stakeholder in Malawi)**

**Programme readiness and logistics**

Feasibility of delivery was another key domain, with ISs emphasising the need to assess EPI capacity including funding, supply chains, cold chain capacity, health worker training, and surveillance infrastructure. In Malawi and Kenya financial sustainability was seen as critical concern, particularly as donor support declines:

*"As a country we are highly donor-dependent so you must consider other critical factors. We need to establish whether we have adequately trained health workers and sufficient infrastructure. When we recommend any vaccine, we need to evaluate if the necessary systems and resources are in place to initiate and sustain this initiative"* **(Institutional Stakeholder in Malawi).**

**3. Trusted sources of evidence**

ISs across the three countries consistently recognised MoHs as the primary custodians of national health data, emphasising that government ownership lends credibility and authority to evidence used in policymaking:

*So as a standard, all the data concerning health is owned by the government. So, data needs to come from the government…any data that you want to use for policy decisions should be data that is hosted by the government. So, because*

*this data is collected usually, through routine health services and these are run by Ministry of Health, and they are sole owners of this data.* **(Institutional Stakeholder in Malawi)**

In addition, research institutions such as the Kenya Medical Research Institute (KEMRI), Malawi-Liverpool Wellcome Trust (MLW), and Addis Ababa University were viewed as trusted sources for high-quality, peer reviewed evidence. International partners, including WHO, UNICEF, GAVI, and other UN agencies, were also cited as credible contributors to the evidence base, as illustrated in Fig 3, trusted sources of evidence for policy decisions.

While CPs did not frame "evidence" in technical terms, they consistently identified health workers, community leaders, and religious figures as their most trusted sources of vaccine information. Their responses reflected the importance of interpersonal trust and local intermediaries in shaping how evidence is understood and accepted. For example, one participant in Malawi explained:

*"I feel like if other vaccines are introduced, people would be happy....... But I would prefer that before any vaccine is introduced, people in the communities should be civic educated about how these vaccines work……the messages can be disseminated using religious leaders because most of the people trust their religious leaders most".* **(FGD participant in Malawi)**

Similarly, Ethiopian participants highlighted frontline health workers as their primary source of knowledge, while also noting frustration when information was incomplete:

*"I just recognised the rota vaccine when you told us it is given in the form of droplets… The community doesn't know the purpose of the vaccine and doesn't know the side effects, and that is because the health workers don't explain the side effects."* **(FGD participant in Ethiopia)**

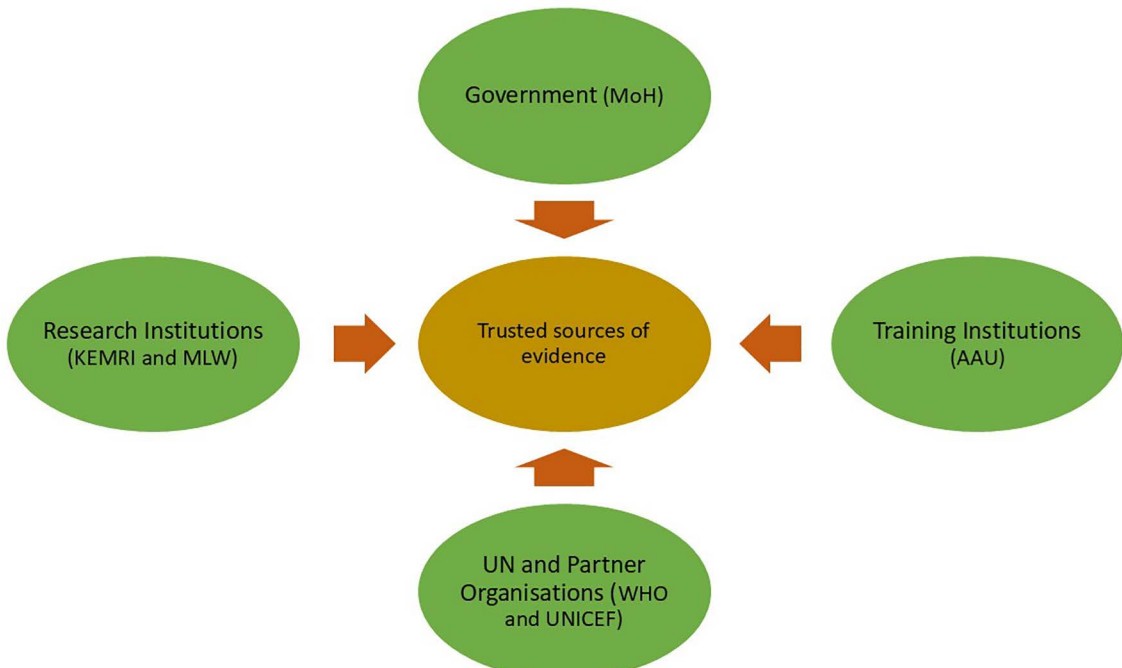

**Fig 3. Trusted sources of evidence for policy decisions.** AAU: Addis Ababa University; KEMRI: Kenya Medical Research Institute; MLW: Malawi-Liverpool-Wellcome Trust; MoH: ministries of health; UNICEF: United Nations Children's Fund; WHO: World Health Organization.

These findings illustrate that while ISs prioritise formal institutional and research sources of evidence, CPs place their trust in proximate, relational sources such as health workers and community leaders. Together, these perspectives highlight the need for strategies that both strengthen institutional data systems and leverage trusted community intermediaries to ensure that evidence is effectively mobilised for vaccine policy and uptake.

**4. Strategies for increasing evidence use**

ISs highlighted that evidence alone is insufficient to inform policy, and deliberate strategies are needed to strengthen its use in decision-making. Several approaches were identified:

**Timely provision of evidence aligned with country needs**

Firstly, ISs emphasised the timely provision of policy-relevant evidence, tailored to different stages of the policy cycle. They noted that evidence must be available early in agenda-setting, during implementation, and in evaluation, ensuring that policymakers have the right information at the right time:

*"So, from the start, data must be available and adequate. Data is needed at different stages and there is different type of data that need to be provided at pre-policy formation stage, another during the implementation, monitoring, etc. So, I would say at all those time points, data is required".* (**Institutional Stakeholder in Malawi**)

**Engaging policymakers in research processes and early involvement of NITAGs**

Secondly, ISs stressed the importance of engaging policymakers and NITAGs early in the research processes to build trust and promote ownership. Policymakers who are involved from the outset are more likely to understand and apply findings, while early NITAG engagement was viewed as key to advocating for vaccine introduction:

*"So, we are also becoming part of that research, for example, so that we can understand the whole process, which builds trust and helps the investigator understand the context. So, most of the time, from my experience, I can say that the evidence produced with the engagement of policymakers and programme people, like me and my expertise, is translated to policy".* (**Institutional Stakeholder in Ethiopia**)

*"What I would suggest is the involvement of NITAG from the beginning, would be key If any new vaccine is to be introduced. If you get NITAG on board, to start discussing these issues about a particular pathogen or the potential for a vaccine, then they help you out to be the advocates and they start looking at evidence for introducing the vaccine early on".* (**Institutional Stakeholder in Malawi**)

**Strengthening National Public Health Institutes (NPHIs)**

Thirdly, ISs underlined the need to strengthen NPHIs and surveillance systems to improve the quality, timeliness, and completeness of data. Reliable national data were considered essential for building confidence and reducing dependence on external sources:

*"Something which has been on the table for many years in Kenya and I think it's the issues of the National Public Health Institute. And I think if established, it will be quite critical in terms of coordinating all these and to make sure that quality data is available for decision makers and that can be relied on when discussions on policy changes and directions need to be done".* (**Institutional Stakeholder in Kenya**)

**Prioritising diarrhoeal diseases and establishing technical working groups (TWGs)**

Fourthly, ISs recommended prioritising diarrhoeal diseases and establishing dedicated TWGs within MoHs. Such groups would provide a specialised platform for reviewing diarrhoeal evidence and complement NITAG deliberations, thereby elevating the policy visibility of diarrhoeal-causing pathogens:

> *"The disease must be recognised as a major public health problem. If it is no more public health problem then it is very difficult to justify, for example, that's what we have as a problem with regards to yellow fever in the country, because the government thinks that it is not a problem".*
>
> **(Institutional Stakeholder in Ethiopia)**

> *"I think having something like a technical working group or coordinating unit just like other major infections like HIV and TB, I think it will help in sort of pushing and making sure that whatever is being done is being translated into actions that can help in alleviating gastrointestinal infections".*
>
> **(Institutional Stakeholder in Malawi)**

**Improving communication for knowledge mobilisation**

Finally, both ISs and CPs emphasised the need for improved communication and knowledge mobilisation strategies. ISs suggested targeted communication channels such as policy briefs, digital platforms, and advisory committees to ensure evidence reaches decision-makers:

> *"There are multiple channels for communicating with stakeholders during the process of making policies, but what would define the best strategy would be segregating the target stakeholders that are being earmarked for influence and then the most appropriate channel would be defined based on that".* **(Institutional Stakeholder in Kenya)**

CPs, meanwhile, highlighted the importance of broader community education campaigns and advocacy strategies delivered through trusted local intermediaries such as health workers and community organisations. However, strategies such as community gatherings, maternal training, radio campaigns, door-to-door outreach need to be tailored to reach vulnerable groups. For example, in Ethiopia, CPs emphasised the importance of targeted follow-ups and advocacy for marginalised populations such as street children:

> *"I have observed that the orphans living on the street don't know what vaccines mean. The youths living there give birth to one on another, and no one follows up on them. We need to support and advocate for them".* **(FGD participant in Ethiopia)**

## Discussion

This study explored institutional stakeholders' and community perspectives on evidence needs and involvement in policy decisions for potential *Shigella* and ETEC vaccine introduction in Ethiopia, Kenya, and Malawi. Vaccine adoption is complex, shaped by political dynamics [37] but requiring robust evidence on disease burden, vaccine profiles, population-specific factors, and implementation feasibility. Key gaps include limited diarrhoeal disease-focused technical working groups and opportunities to strengthen evidence use and community engagement for inclusive policymaking. Disease burden data are critical for guiding intervention strategies, including vaccine adoption, while evidence on vaccine

efficacy, safety, and cost-effectiveness is equally vital to justify prioritisation. Early availability of both datasets is key to accelerating policy decisions and enabling timely action, a principle well-supported by literature on vaccine introduction in LMICs [38,39].

This study highlights the critical role of population-based evidence and the need for accurate, consistent metrics to capture the true population picture for planning new vaccine introductions. Immunisation coverage reflects a program's capacity to integrate new vaccines [40], but its reliability depends on context-appropriate denominators. Standardising data sources and improving accuracy are therefore essential for optimising resource allocation and program impact [41,42]. Programmatic readiness also hinges on a country's political commitment to vaccine adoption and sustainability [43]. For non-GAVI-supported countries, securing independent funding and delivery systems is particularly important. South Africa, for example, was the first African nation to introduce rotavirus and pneumococcal conjugate vaccines, demonstrating how such challenges can be addressed without external support [44]. Beyond financing, programme readiness also requires sufficient staff capacity, functional cold chain systems, and robust surveillance to ensure successful rollout.

Our findings support La Torre *et al.'s* [45] recommendation for Health Technology Assessment (HTA) in vaccine introduction, which aligns with WHO and Burchett frameworks in emphasising multidisciplinary evidence [46,47]. We further validate Donadel *et al*. [38] on the critical need to consider programmatic factors, particularly in LMICs where resource allocation determines implementation success. Evidence from the International Centre for Diarrhoeal Diseases Research, Bangladesh (icddr,b) illustrate these principles in practice. For example, icddr,b rotavirus vaccine trials demonstrated the feasibility, safety, and efficacy of vaccines in resource-limited settings [48], while studies on oral cholera vaccines highlighted the importance of delivery strategies, affordability, and community engagement for successful implementation [26]. These examples demonstrate how robust local research can meet both scientific and programmatic evidence needs, addressing a historical gap in decision-making frameworks that often overlooked programmatic readiness despite its central role in vaccine rollout [23].

Robust local evidence from credible research institutions to inform vaccine policy is critical as highlighted in our study. Yet the absence of local evidence often delays or prevents vaccine introduction, as seen in Indonesia's rejection of the H1N1 vaccine due to limited local data, despite WHO-documented global burden [49,50]. To overcome such barriers, LMICs should strengthen research capacity by enhancing surveillance systems, updating national research agendas, and securing dedicated research funding for multidisciplinary studies. The experience of icddr,b illustrates how sustained institutional investment in diarrhoeal research within an LMIC can yield evidence with profound global policy impact, for example, its pioneering development of oral rehydration solutions (ORS) in the 1960s, which drastically reduced childhood mortality from diarrhoea [25].

Diarrhoeal diseases are among the top five causes of morbidity and mortality in sub-Saharan Africa, exceeding HIV and tuberculosis in Disability-Adjusted Life Years (DALYs) [51]. Given their significant health burden, this study recommends prioritising them in public health by establishing diarrhoeal disease-specific technical working groups within Ministries of Health. Furthermore, integrating community involvement in these efforts can enhance advocacy for incorporating new vaccines into national immunisation programmes.

Our FGDs with CEI groups highlighted the need for clear, detailed vaccine information covering mechanisms, safety, side effects, schedules, and administration. Effective dissemination through trusted sources such as health workers and community leaders (e.g., chiefs, religious figures) was key to fostering parental trust aligning with findings from Ames *et al*. [52] and Oku *et al.* [53]. Additionally, meaningful community engagement in policy processes can enhance ownership, yet contextual reviews are needed to evaluate its actual impact. For example, a Malawi study found that while community involvement is often mandated, stakeholders viewed it as tokenistic, with their input rarely influencing decisions due to power imbalances [54]. Similar evidence across Africa support the need to address barriers and leverage facilitators for effective participation [55].

### Future research

By creating awareness and demand, this work supports the development of potential *Shigella* and ETEC vaccines, which are currently in progress. Although the timeline for licensed vaccines remains uncertain, several candidates are advancing through preclinical and Phase 2 trials—including a recently concluded ETEC trial in The Gambia and an anticipated *Shigella* trial completion in 2025 [56–58]. Regulatory approvals and other factors will influence their availability, but the current progress is promising for near-future deployment.

Key research priorities include generating local *Shigella* and ETEC burden data (e.g., through the Enterics for Global Health *Shigella* surveillance study [59]), evaluating vaccine cost-effectiveness, and exploring community perspectives on diarrheal management and vaccine acceptance. Programmatic readiness, especially in Kenya's post-GAVI transition, also warrants attention. Additionally, studies on community engagement in vaccine policy development are needed to strengthen ownership, acceptance, and alignment of new vaccine initiatives with community needs.

### Strengths and limitations

This study employed a rigorous qualitative approach, using FGDs and IDIs to generate in-depth insights from institutional stakeholders and community representatives across three countries. These methods enabled a nuanced understanding of evidence needs for policymaking and the dynamics of community engagement in health policy processes. By capturing diverse stakeholder voices, the study highlights key factors influencing vaccine policy development and implementation in LMICs.

Several limitations should be noted. First, the purposive sampling strategy, while suitable for the study's aims, limits generalisability and may have excluded other relevant perspectives. Second, conducting only one FGD per country constrained the breadth of community viewpoints; additional FGDs with varied community or civil society groups could have enriched the findings. Third, language translation and transcription may have led to the loss of subtle meanings or cultural nuances, despite measures to preserve the accuracy of participants' responses. Finally, as FGDs were conducted only in urban settings, the findings may not fully reflect rural experiences, where disease burden and access to immunisation services differ.

Nevertheless, the study offers valuable insights into the evidence and engagement processes necessary to support *Shigella a*nd ETEC vaccine introduction. Future research should broaden the range of community perspectives and stakeholder groups to build on these findings and specifically engage caregivers of young children to capture household-level knowledge, attitudes, and perspectives on new vaccines, insights that are critical for informing strategies to strengthen acceptance and uptake.

### Conclusion

This study demonstrates that both ISs and CPs recognise the importance of robust evidence and meaningful engagement in shaping vaccine policies. ISs highlighted four domains of evidence as key to new vaccine introduction including disease burden, vaccine characteristics, and programmatic readiness. However, they identified persistent challenges including limited local data, weak surveillance systems which constrain timely evidence use. CPs, while less involved in national policy processes, emphasised the need for transparent communication about vaccine characteristics particularly safety, side effects, and administration, as essential for acceptance and trust.

These findings highlight the importance of investing in local research capacity, surveillance, and translation mechanisms to ensure that evidence informs policy decisions. Lessons from icddr,b in Bangladesh illustrate how long-term investment in diarrhoeal disease research can yield global policy impact. Building similar capacity in Africa LMICs will be critical introducing new vaccines for diarrhoeal infections such as *Shigella*.

## Policy and practice implications

- Governments and partners should prioritise investments in generating localised evidence for vaccine policy decisions.

- Mechanisms to institutionalise meaningful community involvement in national policy processes must be developed to foster trust and ownership.

- Sustained efforts to improve vaccine communication strategies are essential, with a focus on addressing misconceptions and leveraging trusted messengers.

- Countries should assess and strengthen their programmatic readiness, particularly considering financial sustainability challenges for new vaccines.

- TWGs for diarrhoeal diseases should be established in MoHs, with community representatives.

## Patient data statement (if required)

This study did not utilise any patient data.

## Information governance statement

A data governance plan was developed for this study to outline the research context and establish specific processes for ensuring effective and secure data management. The approach was structured around five key principles: safe projects, safe people, safe settings, safe data, and safe outputs.

At each level, potential data governance risks were identified, and appropriate controls were specified to mitigate risks and uphold data integrity, confidentiality, and ethical standards throughout the study.

## Supporting information

**S1 Text. Topic guides for both Institutional Stakeholders' interviews and focus group discussions.**
(PDF)

**S1 Checklist. Consolidated criteria for Reporting Qualitative research (COREQ) Checklist.**
(PDF)

**S2 Checklist. Inclusivity in global research.**
(PDF)

## Acknowledgments

We would like to express our sincere gratitude to all the participants in this study for their time and valuable contributions to the discussions. We also acknowledge the valuable contribution of the following members of the GHRG-GI consortium: Daniel Asrat (Addis Ababa University, Addis Ababa, Ethiopia); Prisca Benedicto (Malawi Liverpool Wellcome Programme, Blantyre, Malawi); Catherine Beavis (University of Liverpool, UK); Christina Bronowski (University of Liverpool, Liverpool, UK); Jobiba Chinkhumba (Kamuzu University of Health Sciences, Blantyre, Malawi); Helen Clough (University of Liverpool, Liverpool, UK); Jen Cornick (University of Liverpool, Liverpool, UK and Malawi Liverpool Wellcome Programme, Blantyre, Malawi); Khuzwayo Jere (University of Liverpool, Liverpool, UK); James Ngumo Karis (Kenya Medical Research Institute, Nairobi, Kenya); Cecilia Mbae (Kenya Medical Research Institute, Nairobi, Kenya); Edson Mwinjiwa (Malawi Liverpool Wellcome Programme, Blantyre, Malawi), Latif Ndeketa (University of Liverpool, Liverpool, UK and Malawi Liverpool Wellcome Programme, Blantyre, Malawi); Phelgona Otieno (Kenya Medical Research Institute, Nairobi, Kenya);

Virginia Pitzer (Yale, New Haven, USA); Yemisrach Shumeye (Addis Ababa University, Addis Ababa, Ethiopia); Abebe Habtamu Tamire (Addis Ababa University, Addis Ababa, Ethiopia); Fred Were (Kenya Medical Research Institute, Nairobi, Kenya); Mengistu Yilma (Addis Ababa University, Addis Ababa, Ethiopia).

## Author contributions

**Conceptualization:** Chikondi A. Mwendera, Chisomo Msefula, Amha Mekasha, Nigel Cunliffe, Deborah Nyirenda, Neil French.

**Data curation:** Chikondi A. Mwendera.

**Formal analysis:** Chikondi A. Mwendera, Mackwellings Phiri, Rahma Osman, Shewit Weldegebriel, Beatrice Ongadi.

**Funding acquisition:** Chisomo Msefula, Amha Mekasha, Samuel Kariuki, Nigel Cunliffe, Neil French.

**Investigation:** Chikondi A. Mwendera, Mackwellings Phiri, Rahma Osman, Shewit Weldegebriel, Beatrice Ongadi.

**Methodology:** Chikondi A. Mwendera, Mackwellings Phiri, Rahma Osman, Shewit Weldegebriel, Beatrice Ongadi, Catherine Beavis, Jessica A. Fleming, Dan Hungerford, Deborah Nyirenda, Neil French.

**Project administration:** Chikondi A. Mwendera, Chisomo Msefula, Amha Mekasha, Deborah Nyirenda, Neil French.

**Resources:** Samuel Kariuki.

**Software:** Chikondi A. Mwendera.

**Supervision:** Jessica A. Fleming, Dan Hungerford, Nigel Cunliffe, Deborah Nyirenda.

**Validation:** Chikondi A. Mwendera, Mackwellings Phiri, Rahma Osman, Shewit Weldegebriel, Beatrice Ongadi.

**Visualization:** Chikondi A. Mwendera.

**Writing – original draft:** Chikondi A. Mwendera, Mackwellings Phiri.

**Writing – review & editing:** Chikondi A. Mwendera, Mackwellings Phiri, Rahma Osman, Shewit Weldegebriel, Beatrice Ongadi, Chisomo Msefula, Amha Mekasha, Samuel Kariuki, Catherine Beavis, Jessica A. Fleming, Dan Hungerford, Nigel Cunliffe, Deborah Nyirenda, Neil French.

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
