## [Decision Letter · Decision Letter 0]

14 Aug 2025

PGPH-D-25-01318

Evidence needs and community involvement in policy decisions for vaccine-preventable diarrhoeal infections among children under the age of five years: Stakeholder engagement in Ethiopia, Kenya, and Malawi

Dear Dr. Mwendera,

Thank you for submitting your manuscript to PLOS Global Public Health. After careful consideration, we feel that it has merit but does not fully meet PLOS Global Public Health’s publication criteria as it currently stands. Therefore, we invite you to submit a revised version of the manuscript that addresses the points raised during the review process.

We look forward to receiving your revised manuscript.

Kind regards,

Rajiv Sarkar

Academic Editor

Journal Requirements:

2. We have amended your Competing Interest statement to comply with journal style. We kindly ask that you double check the statement and let us know if anything is incorrect.

Additional Editor Comments (if provided):

Reviewer 1 has provided several valuable suggestions, both within the manuscript and in her written comments, including recommendation for referencing relevant work on diarrheal diseases conducted by ICDDR,B. Kindly consider citing appopriate publications from Bangladesh and other LMICs (outside of Africa), as applicable.

Reviewers' comments:

Reviewer's Responses to Questions

**Comments to the Author**

1. Does this manuscript meet PLOS Global Public Health’s publication criteria?

Reviewer #1: Partly

Reviewer #2: Yes

2. Has the statistical analysis been performed appropriately and rigorously?

Reviewer #1: N/A

Reviewer #2: N/A

3. Have the authors made all data underlying the findings in their manuscript fully available (please refer to the Data Availability Statement at the start of the manuscript PDF file)?

Reviewer #1: Yes

Reviewer #2: Yes

4. Is the manuscript presented in an intelligible fashion and written in standard English?

Reviewer #1: Yes

Reviewer #2: Yes

Reviewer #1: The methodology section should be clearer and rewritten. The literature review should be more in-depth. International Centre for Diarrhoeal Disease Research, Bangladesh (icddr,b) has been doing diarrhoeal diseases research for over 6 decades and there is no mention or reference to the research work of this organization. If the result section is enriched with further edits, the Conclusion will take a new shape. In terms of data availability - the guidelines are there.

Reviewer #2: *Can you expand more on the study design, why phenominological?

*The selection of the settings of the FGD's is not optimal, urban areas like AA and Nairobi have significantly higher access to immunization services that their rural counterparts, also less likely to suffer from diarrheal diseases and its complications. This should be highlighted as a limitation.

*Under emerging themes, the first theme is "policy involvement and evidence-related challenges" and them its titled "policy involvement", which one is it?

*The quotes of theme 1 are focused on the roles of the participants in immunization programs nothing significantly related to "evidence related challenges was captured" except issues raised by a respondent in line number 196-198.

*On theme 3...are there any FGD participants that had ideas on who would be the trusted sources of evidence for the community ?

**Do you want your identity to be public for this peer review?** For information about this choice, including consent withdrawal, please see our Privacy Policy

Reviewer #1: **Yes: ** Rasheda Khan

Reviewer #2: **Yes: ** Samuel Z Kidane

---

## [Decision Letter · Decision Letter 1]

10 Nov 2025

Evidence needs and community involvement in policy decisions for vaccine-preventable diarrhoeal infections among children under the age of five years: Stakeholder engagement in Ethiopia, Kenya, and Malawi

PGPH-D-25-01318R1

Dear Dr Mwendera,

We are pleased to inform you that your manuscript 'Evidence needs and community involvement in policy decisions for vaccine-preventable diarrhoeal infections among children under the age of five years: Stakeholder engagement in Ethiopia, Kenya, and Malawi' has been provisionally accepted for publication in PLOS Global Public Health.

Best regards,

Rajiv Sarkar

Academic Editor

Reviewer Comments (if any, and for reference):

Reviewer's Responses to Questions

**Comments to the Author**

Reviewer #1: All comments have been addressed

Reviewer #2: All comments have been addressed

publication criteria?

Reviewer #1: Yes

Reviewer #2: Yes

3. Has the statistical analysis been performed appropriately and rigorously?

Reviewer #1: Yes

Reviewer #2: Yes

4. Have the authors made all data underlying the findings in their manuscript fully available (please refer to the Data Availability Statement at the start of the manuscript PDF file)?

Reviewer #1: Yes

Reviewer #2: Yes

5. Is the manuscript presented in an intelligible fashion and written in standard English?

Reviewer #1: Yes

Reviewer #2: Yes

Reviewer #1: The authors need to further communicate with PLOS data policy about sharing the data. To my understanding, the explanation the author provided is acceptable. However, the author may mention that any set of data will be readily available if it deemed important.

Reviewer #2: N/A

**Do you want your identity to be public for this peer review?** For information about this choice, including consent withdrawal, please see our Privacy Policy

Reviewer #1: **Yes: ** Rasheda Khan

Reviewer #2: **Yes: ** Samuel Z Kidane
